# Anticancer and Antiviral Properties of Cardiac Glycosides: A Review to Explore the Mechanism of Actions

**DOI:** 10.3390/molecules25163596

**Published:** 2020-08-07

**Authors:** Dhanasekhar Reddy, Ranjith Kumavath, Debmalya Barh, Vasco Azevedo, Preetam Ghosh

**Affiliations:** 1Department of Genomic Science, School of Biological Sciences, University of Kerala, Tejaswini Hills, Periya (P.O), Kasaragod, Kerala 671320, India; dhanasvims@gmail.com; 2Centre for Genomics and Applied Gene Technology, Institute of Integrative Omics and Applied Biotechnology (IIOAB), Nonakuri, Purba Medinipur WB-721172, India; dr.barh@gmail.com; 3Laboratório de Genética Celular e Molecular, Departamento de Biologia Geral, Instituto de Ciências Biológicas, Universidade Federal deMinas Gerais (UFMG), Minas Gerais, Belo Horizonte 31270-901, Brazil; vascoariston@gmail.com; 4Department of Computer Science, Virginia Commonwealth University, Richmond, VA 23284, USA; preetam.ghosh@gmail.com

**Keywords:** cardiac glycosides, anticancer, antiviral, viral protein translation, signaling pathway, autophagy, preclinical trials, biomolecules

## Abstract

Cardiac glycosides (CGs) have a long history of treating cardiac diseases. However, recent reports have suggested that CGs also possess anticancer and antiviral activities. The primary mechanism of action of these anticancer agents is by suppressing the Na^+^/k^+^-ATPase by decreasing the intracellular K^+^ and increasing the Na^+^ and Ca^2+^. Additionally, CGs were known to act as inhibitors of IL8 production, DNA topoisomerase I and II, anoikis prevention and suppression of several target genes responsible for the inhibition of cancer cell proliferation. Moreover, CGs were reported to be effective against several DNA and RNA viral species such as influenza, human cytomegalovirus, herpes simplex virus, coronavirus, tick-borne encephalitis (TBE) virus and Ebola virus. CGs were reported to suppress the HIV-1 gene expression, viral protein translation and alters viral pre-mRNA splicing to inhibit the viral replication. To date, four CGs (Anvirzel, UNBS1450, PBI05204 and digoxin) were in clinical trials for their anticancer activity. This review encapsulates the current knowledge about CGs as anticancer and antiviral drugs in isolation and in combination with some other drugs to enhance their efficiency. Further studies of this class of biomolecules are necessary to determine their possible inhibitory role in cancer and viral diseases.

## 1. Introduction

Global development of cancer registries has led to the discovery of novel drugs that have been derived from natural sources that are used to treat several maladies including cancers and many viral diseases [1]. Several cancers including, breast, lung, liver, colon, gastric, glioblastoma, acute myeloid leukemia and pancreatic cancers pose a constant threat to human health due to the lack of effective therapeutic options, and this incidence is expected to rise by 70% in the next two decades [2]. In addition to these, several viral infections including both DNA (Cytomegalovirus (CMV), herpes simplex virus and Adenovirus) and RNA (chikungunya virus, coronavirus, respiratory syncytial virus, Ebola virus, influenza virus and human immunodeficiency virus (HIV) have also threatened world health due to a similar lack of target-specific therapeutic agents [3]. The search of novel therapeutic agents for anticancer and antiviral diseases has demonstrated that plants remain the biggest reservoir for novel drugs and provides a promising line for research on the aforementioned diseases [4].

About 80% of the world’s population depends on traditional medicines for several diseases including anticancer and antiviral treatments. More than 60% of the clinically approved drugs for cancer therapy are from the plant sources and around 20–30% of the drugs available in the market were obtained from plants [5]. A study by Newman and Cragg has demonstrated that, 16% of biologic macromolecules (*n* = 250), 4% of unaltered natural products (*n* = 67), 1% of botanical drugs (defined mixture) (*n* = 9), 21% of natural product derivatives (*n* = 320), 27% of synthetic drugs (*n* = 420), 11% Synthetic drug (NP pharmacophore)/mimics of natural products (*n* = 172), 10% of synthetic drugs (natural product pharmacophores) (*n* = 162), 6% of vaccines (*n* = 101) and 11% of mimics of natural (*n* = 172) products have been approved from 1981–2014 (*n* = 1562). Among those, 246 (*n* = 246) were approved to treat several cancers. Out of these 246, 14% of biologic macromolecules (*n* = 33), 12% of unaltered natural products (*n* = 30), <1% of botanical drugs (defined mixture) (*n* = 1), 25% of natural product derivatives (*n* = 62), 19% of synthetic drugs (*n* = 47), 10% Synthetic drug (NP pharmacophore)/mimics of natural products (*n* = 24), 9% of synthetic drugs (natural product pharmacophores) (*n* = 22), 2% of vaccines (*n* = 5) and 9% of mimics of natural products (*n* = 22) were reported [6]. Cardiac glycosides (CGs) are among the natural compounds whose native function is to inhibit Na^+^/K^+^ ATPase (sodium-potassium pump) activity, which is an universal enzyme responsible for translocating Na^+^ and K^+^ ions transversely on the cell membrane using ATP as the energetic force to maintain intracellular ion homeostasis and to create the positive inotropic effect in heart failures. Recent reports on Na^+^/K^+^ ATPase suggested that this ubiquitous enzyme also acts as a classic receptor for several signaling cascades responsible for cell death, cell proliferation and other cellular events.

Moreover, CGs act as inhibitors of cancer cell proliferation and induces apoptosis in several solid cancer malignancies. These in vitro data were supported by the epidemiological studies stating that patients under treatment with CGs were protected from various cancers. Subsequently, CGs have developed as potential antiviral agents by targeting cell host proteins to reduce resistance to antiviral therapies, making them a very promising approach against human viral infections [3]. Interestingly, some of the CGs (digitoxin and digoxin) also possess anti-inflammatory activity by an on-target mechanism. Certainly, these CGs increase the intracellular Na^+^ that further inhibits the ATPase activity of the RNA sensor RIG-1, which is responsible for the activation of signaling pathways and ultimately leads to the secretion of interferon β [7].

### 1.1. CGs as Therapeutic Candidates to Treat Cancers

To date, cancer is one of the foremost causes of death globally. Due to extensive research from the past decade, researchers have investigated several targets to treat cancers with an enormous number of small molecules/drugs. However, these were still not adequate to treat the disease. In this regard, CGs have emerged as one of the promising candidates to treat different types of solid tumors such as lung, breast, liver, colon, gastric, glioblastoma, acute myeloid leukemia, prostate and pancreatic cancers.

#### 1.1.1. Lung Cancer

Lung cancer is the prominent reason for cancer-related mortality around the world. Some biomarkers have been tested to treat this cancer, but the success was limited because of the fundamental and acquired resistance for the particular targets of such drugs [8]. CGs are one of the emerging classes of drugs that possess potential anticancer activities against lung cancers at relatively low concentrations [9,10]. Several CGs such as peruvoside [11], strophanthidin [12], lanatoside C [13], digitoxin, digoxin, convallatoxin, ouabain and glucoevatromonoside [14] have been reported for the lung cancer domain. Correspondingly, several mechanisms were also reported for such CGs to inhibit the growth and proliferation of lung cancers. Lung cancer is known to express high levels of the α-1 subunit of Na^+^/K^+^ -ATPase along with glioma, melanoma and other renal cancers. The overexpression of this subunit does play crucial roles by altering several signaling pathways to cause apoptosis, autophagy and other cell death mechanisms in humans because of the higher sensitivity of the α-1 subunit with CGs compared to rodents and mice systems [15]. For instance, the underexpressed p53 in lung cancer cells showed that CGs can cause cell death through cell cycle arrest at G0/G1 in p53-positive H460 cells [8].

In this regard, targeting lung cancer through the α-1 subunit of Na^+^/K^+^ -ATPase could be a promising approach to treat this disease. Na^+^/K^+^ -ATPase was earlier investigated for treating STK11 mutated lung cancer cells by using CGs as therapeutic drugs. STK11 mutation is considered to be the major mutation for lung cancer progression, and direct therapy was not implemented due to the loss of function of STK11. On the other hand, several CGs such as digoxin, digitoxin and ouabain were known to inhibit lung cancer progression by hindering the expression of α-1 subunit and exhibited discriminatory antitumor effects in STK11 mutant lung cancer cells. Hence, the STK11 mutation may serve as a novel biomarker for treating lung cancers for CGs [16].

#### 1.1.2. Breast Cancer

CGs were considered as the phytoestrogens and hence, it has been linked to the risk of breast and uterus cancers. The connotation between CGs and uterus cancers is well known, whereas the link between breast cancers and CGs was not elucidated [17]. Breast cancers are the most prevalent cancer in women and are second only to lung cancer in cancer-related deaths worldwide [18]. It is a heterogeneous disease, known to possess several subtypes based on the presence/absence of several receptors such as human epidermal growth factor receptor 2 (HER2). However, triple-negative breast cancers (TNBCs) do not express any of the receptors and consequently are resilient to beleaguered therapies [19]. In this regard, CGs have emerged as probable anticancer agents that impart their anticancer activity by targeting several signaling pathways to inhibit breast cancer proliferation. CGs were known to be more sensitive to breast cancers since Na^+^/K^+^ -ATPase is a key player for several cellular functions and acts as a signal transducer for several hormones including estrogens. Moreover, the aberrant expression of this enzyme would lead to the progression of breast cancers [20]. Recently, several research groups have suggested that CGs have the potential to inhibit breast cancer proliferation and is selective to only malignant cells. For instance, digitoxin, digoxin, peruvoside, strophanthidin, ouabain, convallatoxin, oleandrin, proscillaridin and lanatoside C were shown to suppress the growth of breast cancers [10,11,12,13].

#### 1.1.3. Liver Cancer

Liver cancer is the second most deadly and fifth-most affected cancer worldwide [21]. The poor prognosis of hepatocellular carcinoma (HCC) reveals the significance of developing therapeutic compounds to treat liver cancers. We highlight the role of CGs as promising anticancer agents to treat liver cancer [22]. Although several studies reported the anticancer activity of CGs, we still could find only a limited number of reports on the role of CGs against liver cancers [23]. For instance, bufalin, a cardiac glycoside derived from *Bufa bufa*, is known to inhibit cancer cell proliferation, migration and adhesion in HCC derived cell lines. The mechanism of action of this cell death includes the decrease in the intracellular pAKT, pGSK3β, MMP9 and MMP-2 levels and increase in E-cadherin and GSK3β protein levels [24]. Notwithstanding, another report suggested that lanatoside C induces apoptosis in HepG2 cell lines by altering the expression of PKCδ [25] and it also induces apoptosis in Mahlavu liver cancer cells by altering the expression patterns of Akt and ERK signaling. ouabain and cinobufagin treatment to HepG2 cells induces apoptosis through the attenuation of ERK signaling and cMyc activation [26]. Neriifolin—a less commonly studied CG—also possesses anticancer activity against HepG2 cells by arresting the cell cycle at S and G2/M cell cycle phase [27]. Recent reports from our lab have suggested that CGs such as peruvoside, strophanthidin and lanatoside C comprehends a broad spectrum of anticancer activity against HepG2 cells. The mechanism of this cell death was identified as the inhibition of proto-oncogenes and cell cycle arrest and by the alteration of several signaling pathways such as PI3K/AKT/mTOR, Wnt/β-catenin signaling, SAPK/JNK signaling and through MAPK signaling [11,12,13].

#### 1.1.4. Colon Cancer

Colon cancer is considered as one of the metabolic cancers and is ranked as the second most prevalent in women and the third most common cancer in men [28]. The most common mutations with KRAS and p53 leads to colon cancer [29]. Due to the high mutational rate of p53 in colorectal cancers, research must continue to discover and identify molecules that are effective in inhibiting the growth of colorectal cancers that lack functional p53. In this regard, CGs have developed as promising candidates to treat colorectal cancers by targeting several signaling machinery. However, none of these effects could correlate with the functional p53. Only a few reports were available that suggest the anticancer effects of CGs against colorectal cancers. convallatoxin induces apoptosis in HCT116 cell lines by inhibiting the expression of PUMA and NOXA. Both of these genes were known to be expressed by several transcriptional factors that include p53 [30].

Another study with the use of glucoevatromonoside, have suggested that CGs may also cause apoptosis by p53 dependent and independent manner in colorectal cancer cells [14]. In addition, this anticancer activity was validated by the caspase-dependent mechanism. Another study by Kang et al. has suggested that lanatoside C inhibits colorectal cancer growth by inducing mitochondrial dysfunction and increased radiation sensitivity by impairing DNA damage repair [31]. oleandrin, strophanthidin, gitoxigenin and convallatoxin were also found to inhibit the growth of colorectal cancers. Among these, oleandrin was found to be more effective compared to other CGS. The mechanism of this effect was associated with the mitochondrial pathway, intracellular stimuli Ca^2+^ overload and the diminishment of antioxidant glutathione levels [32].

#### 1.1.5. Gastric Cancer

Gastrointestinal cancer accounts for more than 30% of all the cancers globally and most of these reports were predominant in the male population [33]. The median survival rate of gastric cancer patients is 12 months and the 5-year survival rate is less than 10%. This indicates the need for developing potent drugs or therapeutic targets to combat gastric cancer [34]. In the current review, we demonstrated the role of CGs against gastric cancers by delineating the possible mechanism of actions. There were very few reports discussing the role of CGs in gastric cancer; bufalin was the only CG reported so far to have the property to inhibit the gastric cancer cell proliferation. bufalin reverts the acquired cisplatin resistance by inhibiting the stemness markers such as CD133, SOX2 and OCT4. Along with that, bufalin also overcomes the delay in the resistant pattern of cisplatin treatment to gastric cancer cells [35]. Alongside, bufalin is also known to possess anti-invasion and anti-metastatic activity against gastric cancer cells by downregulating the Wnt/β-catenin signaling pathway along with consequent inhibition of ASCL2 and EMT expression [36].

#### 1.1.6. Glioblastoma

Glioblastoma is one of the highly aggressive forms of cancers in the world. Due to its highly aggressive and invasive nature, the prognosis for this type of cancers remains unclear until today. Glioblastoma relapses even after the treatment with surgery, radiation and chemotherapy with a median survival of 14 months [37]. One reason for the relapse in post-treatment could be the presence of glioma stem cells (GSC), which directly acts as the source point for tumor initiation and are supposed to contribute to the resistance to conformist therapies [38]. The development of novel chemotherapeutic agents that can effectively target the GSCs could be a novel approach to treat glioblastoma. All these GSCs are augmented in the perivascular position and expanses near necrosis [39], which in turn are associated with hypoxia. Due to the highly lethal nature of GBM, new therapies are urgently needed, and repositioning of existing drugs also could be a promising approach for developing such therapies. In this scenario, CGs have appeared as one of the most promising approaches as these compounds are known to target HIFs to induce apoptosis in cancers [40]. In particular, a study conducted by Lee et al. has demonstrated that digoxin particularly targets HIF-1α in human glioma stem cells to induce apoptotic effects in glioblastoma [38]. As the GSCs play a main role in the cancer stemness and progression, targeting this with CGs may be a promising approach to exterminate tumor formation. Apart from targeting the HIFs, CGs also induces apoptosis in glioblastoma through the activation of GSK3β and by the alteration of microtubule dynamics. Proscillaridin A acts as the activator for GSK3β by reducing the EB1 comet length and inhibition of glioblastoma migration at relatively less concentration [41].

#### 1.1.7. Acute Myeloid Leukemia

Notwithstanding the advancements in cancer disease detection and drug development, Acute myeloid leukemia (AML) remains as one of the difficult diseases to cure due to its continuous relapse post-treatment. AML is characterized by the overproduction of immature white blood cells, and the shreds of evidence have suggested that the formation of genetic lesions at the hematopoietic stem cells results in the formation of AML [42]. These leukemic stem cells (LSCs) are believed to be the reason for the relapse in the post-treatment and also the reason for developing the resistance to chemotherapeutic drugs [43]. Hence, the development of more efficient treatments with novel targets that specifically target LSCs and probably cure AML is still much needed. ouabain and digitoxin were demonstrated to treat human leukemia. However, the output was not adequate due to the use of high concentrations. In this regard, peruvoside was found to be more promising than that of ouabain and digitoxin causing cell death at very low concentration in human primitive AML cells (KG1a) and chronic myelogenous leukemia (CML) K562 cells. peruvoside showed relatively very low concentrations for both the cell types and induced the highest degree of apoptosis [44]. Evidence has also shown that ouabain induces apoptosis in AML by targeting CD34^+^CD38^−^ [45]. The apoptosis by peruvoside in AML is due to the cell cycle arrest at the G2/M phase, which is one of the basic properties of CGs to induce apoptosis. Along with that, peruvoside also induced apoptosis by triggering the cleavage of Caspase 3, 8 and PARP in KG1a cells. On the other hand, apoptosis was also induced by the upregulation of the CDKN1A mRNA levels but did not cause any changes in other pro-survival gene expressions (SURVIVIN and BCL2). Another study by Hallböök et al. has suggested that digitoxin appeared to be more cytotoxic for primary B-precursor and T-ALL cells [46].

#### 1.1.8. Prostate Cancer

Prostate cancer is one of the common malignancies worldwide and its prevalence has increased rapidly in the last decades [47]. Metastatic prostate cancer is one of the deadliest forms of prostate cancers which is effectively controlled in an androgen-dependent and independent manner. An extensive search for the identification of potential drug candidates has resulted in the investigation of CGs as promising candidates [48]. Johnson et al. have first shown that CGs (ouabain and digitoxin) induce apoptosis in human prostate cancer cell lines (PC3) by inhibiting the expression of Hoxb-13, hPSE/PDEF, hepatocyte nuclear factor-3α and SURVIVIN [49]. Along with that, Digitalis can also inhibit the proliferation of androgen-independent prostate cancer cells (DU and PC3) and androgen-dependent cancer cell lines (LNCaP) by enhancing the accumulation of intracellular Ca^2+^ and by apoptosis [50]. Conferring to the study conducted by Huang et al. [51], ouabain inhibits the growth of androgen-independent prostate cancer cells at nanomolar concentrations which is almost equal to the therapeutic plasma concentrations. Anvirzel has shown promising results in cell apoptosis in human PC3 and C4-2 cells. The mechanism behind this cell death was identified as the fact that Anvirzel significantly inhibited the length of telomeric DNA and also arrested the cell cycle at the G2/M phase [52]. Newman et al. have demonstrated that oleandrin inhibits the expression of FGF-2 and also acts as a potent inhibitor for NF-kB from PC3 and DU145 cells in time and dose-dependent manner [53,54]. Nonlethal or chronic low doses of digitoxin, digoxin and ouabain inhibit the expression of the PSA gene by altering PDEF gene expression in human prostate cancer cell lines (LNCaP) [55]. Along with that digitoxin inhibits the expression of HIF-1α synthesis and blocks tumor growth in prostate cancer [40]. A systematic screen out of 2000 drugs has revealed that five CGs (ouabain, peruvoside, digoxin, digitoxin and strophanthidin) effectively induce cell death in anoikis resistant PP-C1 prostate cancer cells. Furthermore, ouabain initiated anoikis through mitochondrial caspase activation by suppressing the effect of Na^+^/K^+^-ATPase [56].

#### 1.1.9. Pancreatic Cancer

Pancreatic cancer is one of the digestive system malignancies with a very low five-year survival percentage (~6%) with an estimated rate of 227,000 annual deaths per year globally [57]. Unfortunately, the clinical symptoms for pancreatic cancer are not eligible for surgical removal [58]. Hence, chemotherapeutics signifies the predominant strategy for pancreatic cancer treatment [59]. The available chemotherapeutic options can treat the modest forms of cancer but the survival rate with continuous treatment was poor for clinical significance [60]. The limited number of available therapies necessitates the need to develop novel therapeutic options to fight against pancreatic cancer. CGs have shown such promising effects by inhibiting the growth of pancreatic cancer in vitro and in vivo. bufalin has shown the antitumor effect against the human pancreatic mice system (BxPC3-luc2) and cancer cells (Sw1990 and BxPC3) by arresting the cell cycle at G0/G1 phase and also suppressed the expression of cyclin D1 and E1 in pancreatic cancer cells [59]. Another study by Newman et al. [61] has shown that oleandrin at nanomolar concentrations inhibits the growth of pancreatic cancer cells (PANC-1) by arresting the cell cycle at G2/M phase in dose and time-dependent manner. Furthermore, this effect was further confirmed by identifying the drug-dependent inhibition of pAkt and overexpression of pERK. Transfection of Akt into PANC-1 cells resulted in the pERK activation and which was further repealed by oleandrin.

## 2. Molecular Targets of Cardiac Glycosides in Cancers

Though the anticancer activity of CGs may seem pleiotropic, a serious review of the literature may nonetheless permit a molecular signature of cell signaling intermediates from CG treatment, which is yet to be identified. Here we postulate, one or several of these intermediates to the best of our knowledge for the understanding of the mechanism of CGs. Some of the overexpressed intermediates would be one of the superficial targets to treat and once inhibited, would automatically lead to cell death. Here, we highlight some of the predominant molecular targets to treat cancers by using CGs as the therapeutic candidates (Table 1) and the possible mechanism of CGs induced cell death is shown in Figure 1A.

## 3. Cardiac Glycosides as Immune Modulators

Retrospective clinical data demonstrated the use of CGs as possible anticancer drugs to treat cancer patients. However, several CGs are known to stimulate the immune response to several diseases including cancers at multiple stages. In accordance with this, CGs activate the immunogenic cell death (ICD) of various cancer cells [100]. Several CGs such as lanatoside C, digoxin, digitoxin and ouabain particularly act as efficient inducers for ICD in vitro. Further, the mechanism of ICD was validated as the ecto-expression of calreticulin expression, HMGB1 release and ATP secretion on human cancer cells and mouse systems. Subsequently, CGs were identified to stimulate the antitumor immune response in vivo by discovering the role against murine colon cancer cells treated in combination with digoxin and chemotherapy [101]. Furthermore, CGs intensified the anti-neoplastic effects by DNA damage in combination with mitomycin c and cisplatin in the immunocompetent mice model. Here the combination of digoxin with mitomycin c resulted in the more pronounced destruction of tumors by interferon γ-producing α/β CD4+ or CD8+ T lymphocytes compared to that in isolation [101]. Apart from this, CGs can also reduce the off-target effects where these CGs bind to the estrogen receptors (ER) because most of the Digitalis compounds are phytoestrogens and tend to bind with ER with low affinity than that of estrogen. Additionally, CGs may also contribute to the antagonistic activity on ER, where digoxin plays a crucial role in the steroid receptors. This finding was further validated at a large scale chemical screen where, digoxin particularly acted as retinoic acid receptor inhibitor [102].

## 4. Role of Cardiac Glycosides on Signaling Pathways for Their Anticancer Mechanism

CGs have been used for decades to treat congestive heart failures and cardiac arrhythmia. Because of the mutations in the Na^+^/k^+^-ATPase, it has been linked with several diseases including diabetes and Alzheimer’s disease and other bipolar diseases. Recent reports have suggested that the mutation in the sodium–potassium pump could lead to cancer cell proliferation. Several signaling pathways were involved in the process of these diseases such as epithelial-to-mesenchymal transition (EMT), p38 mitogen-activated protein kinase (MAPK) cascade, PI3K/Akt/mTOR (PAM) signaling, p21 Cip and cholesterol homeostasis. Interestingly, all these pathways are known for cancer promotion and are linked to α and β subunits of Na^+^/k^+^-ATPase [103]. Out of these, β subunit plays a crucial role in cancer suppression by tumorigenesis and cancer metastasis. On the other hand, methylation of ATP1B1 inhibits the activity of the β subunit and encourages cancer growth in renal cell carcinoma [104].

### 4.1. Effect on EMT

Epithelial-to-mesenchymal transition is a process, where the epithelial cell changes their phenotype to acquire mesenchymal properties to increase the migrative ability required for cancer progression and invasion all over the body [105]. The β subunit of sodium–potassium pump plays a major role in this process and regulates the integrity of cell polarization [104]. During this process, β-subunit dimerizes with nearby β subunit to increase the cell-to-cell adhesion by forming β-β subunit bridges. During this course of time, the expression of the β subunit decreases along with E-cadherin, which is responsible for the EMT and in the process of cell invasion. This decreased E-cadherin results in the increased activity of β-catenin, which ultimately promotes cancer metastasis [106]. Snail, Zinc finger protein SNAI1, plays a major role in the inhibition of E-cadherin and this snai1 is also responsible for the suppression of β-subunit of the Na^+^/k^+^-ATPase in cancer cells [107].

### 4.2. Effects on p38 MAPK/ERK Signaling Pathway

The MAPK pathway proteins are known to play a crucial role in cell survival, cell cycle and cell death. Proteins such as c-Jun, JNK, MEK1/2, ERK1/2 and p38MAPK play a major role in these cellular events [108,109]. Reports have suggested that the inhibition of Na^+^/k^+^-ATPase regulates the MAPK pathway and leads to cell death and cell cycle arrest. Ye et al. has shown that the inhibition of Na^+^/k^+^-ATPase leads to the interaction between v-src avian sarcoma (Schmidt-Ruppin A-2) viral oncogene homolog (Src) kinase and ultimately attenuates MAPK/ERK pathway [110]. In addition, the ATP/ADP ratio is responsible for the autophosphorylation of Src kinase [111]. ouabain activates p38MAPK by inhibiting the activity of Na^+^/k^+^-ATPase [112] and this activation promotes the transcription of p53 and NF-kB. Thus, activated NF-kB can trigger the Fas-mediated apoptosis of cancer cells.

### 4.3. Effects on Src Kinase Signaling

Src is a non-receptor protein tyrosine kinase that functions to promote cancer cell proliferation and invasion [113]. Inhibition of Na^+^/k^+^-ATPase with CGs leads to the activation of Src which in turn interrelates with EGFR to promote a signaling cascade of Ras to MAPK [114]. The increased activity of Src promotes cell survival and enhances ROS production. Furthermore, this activity of Src could be suppressed by the tyrosine residue on the α-subunit of the Na^+^/k^+^-ATPase [111]. This hindering effect promotes the activation of Src and helps in the proliferation [71].

### 4.4. Effects on PI3K/Akt/mTOR Pathway

PAM signaling is known for its diverse cellular functions such as cell survival, cell death and autophagy [113]. This cascade consists of three important proteins phosphoinositide 3-kinase (PI3K)/protein kinase B (Akt)/the mammalian target of rapamycin (mTOR). Among these, PI3K acts as the regulator protein that activates with the α-subunit of Na^+^/k^+^-ATPase. Binding of proline-rich domain on the Na^+^/k^+^-ATPase with the regulatory subunit p85 promotes the activity of PI3K [115]. This modulation on Na^+^/k^+^-ATPase leads to the autophagic pathway activation and ultimately promotes cancer cell survival [116]. Activated PI3K phosphorylates phosphatidylinositol 4,5-bisphosphate to phosphatidylinositol (3,4,5)-trisphosphate and is responsible for the phosphorylation/activation of Akt. Activated Akt can stimulate the activity of mTOR and promotes the processes of cell survival, proliferation and evasion of autophagy. The mTOR is typical serine/threonine kinase having two subdomains, namely mTOR complex 1 and mTOR complex 2 which is responsible for autophagic cell death and cellular homeostasis [117].

### 4.5. Effects on Autophagy

Autophagy is a process where non-essential or abnormal cells will be engulfed to maintain cell clearance through the process of programmed cell death [118]. It has been reported that CGs can either induce or inhibit autophagy by suppressing Na^+^/k^+^-ATPase in cancer cells. Apart from this, CGs (especially ouabain) can also inhibit autosis (a form of autophagy without apoptosis and necrotic features) in cancer cells [119]. Moreover, ouabain also sensitizes drug-resistant glioblastoma cells to necroptosis by upregulating ATP1A2 and ATP1A3 [120]. Additionally, ouabain also induces autophagy by JNK dependent decrease of Bcl-2 in A549 and H1975 cells [74]. Another study found that digoxin and ouabain induce autophagy by altering mTOR and ERK1/2 through pathway crosstalk mechanism in NSCLC cells [118].

## 5. Effects of Cardiac Glycosides on Gene Expression and Other Pathways

Apart from this, CGs can also trigger several other genes that are responsible for diverse cellular functions. For instance, digoxin and proscillardin A inhibits DNA topoisomerase I whereas bufalin and digoxin inhibit DNA topoisomerase II as well [66,70,88] to induce cell death in cancer cells. Moreover, some CGs inhibits TNF-α along with NF-kB and c-Myc to induce cell cycle arrest [121]. Furthermore, inhibition of Na^+^/k^+^-ATPase also leads to the suppression of several resistant proteins, which allows cancer cells to resist against chemotherapeutic drugs [122]. peruvoside, strophanthidin and lanatoside C show caspase-dependent apoptosis in human breast, lung and liver cancer cells to induce mitochondrial cell death [11,12,13].

## 6. Cardiac Glycosides in Clinical Trials for Cancer Therapy

CGs have a long history in treating heart diseases however, recent clinical trials on phases I and II have proved that these compounds also possess anticancer activities against various solid tumors. Three CGs have been reported to be in the preclinical trials to determine the dose-limiting toxicities and to identify the maximum tolerated dose (MTD) limit for its usage as anticancer drugs [9]. However, there exists a paucity in the animal studies for the CGs due to the infrequent species-dependent compassion to the inhibition of cancer cell proliferation. This growth inhibitory effect was reported in the year 1967 [123]. In the past decade, there has been an extensive increase in animal studies to identify the possible mechanism of apoptosis in cancer cells upon CGs treatment.

Several CGs were included in clinical trials, among which Anvirzel (an aqueous extract from *Nerium oleander*), followed by PBI-02504 (CO_2_ extract of *Nerium oleander*) and UNBS-1450 (a semisynthetic derivative of 2″-oxovuscharin extracted from *Calotropis procera*) and digoxin were in phase I clinical trials [124]. Anivirzel is similar to oleandrin, neritaloside and oleandrinigen and its function is to inhibit the FGF-2 in prostate cancer cells in time and dose-dependent manner to induce cell death [53]. The first phase I clinical trial was started by Mekhali et al. in 2006 to identify the MTD and drug safety in 18 patients with cutting-edge refractory tumors [125]. The results of the phase I clinical trials were promising and can safely be administrated intramuscularly by up to 1.2 mL/m2/day. Apart from this role as an anticancer agent, Anvirzel was also in phase I clinical trials, tested for its activity against NSCLC alone and in combination with several other drugs such as carboplatin and docetaxel [124].

Fascinatingly, PBI-05,204 also contains oleandrin in its core structure whose function is to inhibit the α-3 subunit of the Na^+^/K^+^-ATPase and also suppresses NF-kB signaling to induce apoptosis. PBI-05,204 was known for its phosphorylating activity of Akt and p70S6 K to alter the activity of mTOR [126]. MTD (0.6–10.2 mg/day) in the phase I clinical trials was satisfactory and recommended for phase II clinical trials for treating several solid cancers such as bladder, colon, rectum, breast and pancreas. The results from the clinical trials exhibited the safety, pharmacokinetics and pharmacodynamics of PBI-05,204 and identified the recommended dose as 0.2255 mg/kg [127]. Presently, PBI-05,204 was in clinical trials to treat solid pancreatic cancers (https://clinicaltrials.gov/ct2/show/NCT02329717).

To date, digitoxin was under preclinical trials for 16 different solid tumors (breast, lung liver, Kaposi’s sarcoma, AML, melanoma, head and neck, etc.) out of which eight have already been completed (https://clinicaltrials.gov/ct2/results?cond=cancer&term=digoxin). Apart from this, digitoxin was also tested for its anticancer activity in combination with several chemo and immunotherapeutic drugs for efficient activity. For instance, head and neck patients treated with digitoxin and cisplatin have shown better results than that of the standalone treatments (https://clinicaltrials.gov/ct2/show/NCT02906800) [128]. digitoxin, has cleared the phase II clinical trial for its usage to treat breast cancer in the year 2020 and also cleared phase II clinical trials for its usage in treating prostate cancer. Apart from this, digitoxin in combination with Tivantinib has cleared the phase 1 clinical trials to treat solid tumors. A Phase 1B clinical trial for digitoxin in combination with Trametinib was completed in the year 2018 to treat patients with unresectable or metastatic BRAF wild-type melanoma (https://clinicaltrials.gov).

UNBS1450 was tested in clinical trials using a dose intensification study to find out the MTD, toxicity and pharmacokinetic parameters against patients with lymphoma. The activity of this compound was tested in preclinical trials for 57 different types of solid tumors [129]. In preclinical trials, UNBS1450 has shown better results than the reference compounds such as paclitaxel irinotecan, oxaliplatin, mitoxantrone and temozolomide [130] for prostrate [131], glioblastoma [132] and NSCLC [84]. The advanced feature of this compound is that it can inhibit three isoforms (α3β1, α2β1 and α1β) with relatively higher efficiency (~6 to >200 times) than ouabain and digoxin [132]. UNBS1450 induces an apoptotic and non-apoptotic form of cell death depending on the cellular atmosphere. Non-apoptotic cell death mechanism such as lysosome membrane permeabilization and autophagy was witnessed in solid tumors, which helps in avoiding the apoptotic resistant pathways. In the apoptotic form of cell death, several distinct features have been shown such as anti-apoptotic proteins Bak and Bax activation. The activation of these proteins leads to the cytochrome c release and caspase cleavage responsible for cell death [133]. Moreover, another study by Orrenius et al. has shown that UNBS1450 can suppress oncoprotein c-Myc and related genes to induce apoptosis cell death [134]. Regrettably, the phase I clinical trials was closed in the year 2011 by the sponsor due to bankruptcy before reaching the MTD from 23 patients.

## 7. Antiviral Activities of Cardiac Glycosides and Their Mechanisms of Action

Apart from the anticancer activities, some of the CGs were also reported to possess antiviral and anti-neoplastic effects. These multiple inhibitory properties of CGs have made them one of the most suitable candidates for the drug repurposing approach. CGs such as digoxin, ouabain, digitoxin and convallatoxin were reported to inhibit Cytomegalovirus, whose function is to paralyze the adaptive immunity. Along with digoxin, ouabain and digitoxin, G-strophanthidin were known to inhibit herpes simplex virus, which functions in degrading the host mRNA [135]. digitoxin and digoxin were also reported to inhibit Adenovirus through sodium-potassium pump inhibition [136]. This Adenovirus is well known for delivering the viral genome inside the cells. Apart from the inhibition of DNA viruses, CGs were also reported to inhibit RNA viruses. digoxin, ouabain and digitoxin were known to inhibit the chikungunya virus and coronaviruses by inhibiting the Src pathway and also by inhibiting viral entry. ouabain was also reported to inhibit the respiratory syncytial virus and the Ebola virus by altering the viral RNA. Recent reports also suggested that CGs such as ouabain, digoxin and lanatoside C were known to inhibit the influenza virus by inhibiting viral protein translation. Digoxin, ouabain, lanatoside C and digitoxin were reported to inhibit the Human immunodeficiency virus by altering the viral pre-RNA splicing [3]. A list of CGs with potent antiviral activities is reported in Table 2 and the illustration is shown in Figure 1B.

## 8. Conclusions and Future Perspectives

Cardiac glycosides have a long history in treating heart diseases, but recent studies on cancer cell lines and animal systems have demonstrated the anticancer and antiviral activities of several CGs. Depending on these findings CGs have been identified as potential anticancer and antiviral agents that should be assessed in clinical studies. Primarily CGs act as targets for Na^+^/K^+^ ATPase, which has a role in attenuating several signaling pathways linked to cell proliferation, apoptosis and autophagy. One interesting fact is that CGs acts on the membrane targets due to their nature in adapting the membrane fluidity. However, there was no clear evidence yet stating the lipid permeability and direct contact with the targets by CGs which needs to be discovered. The key feature of any drug compound is that it should act in a target-specific manner and should be active at very less concentration that is usually nontoxic to other cells and has the chance of being used in the clinical studies. Based on this principle, CGs have shown the anticancer activity at nanomolar concentration against various cancer cells and antiviral activity on several viral diseases.

Within a narrow time window, several CGs have been developed for clinical trials (Anvirzel, UNBS1450, PBI05204 and digitoxin) for their anticancer activities against solid tumors and some of them were FDA approved for their activities against heart diseases (digitoxin, digoxin and lanatoside C). Regarding their anticancer activities, several recommendations have been made for evaluating their anticancer potential. The anticancer and antiviral activities of CGs and their molecular targets have been discussed in an increasing number of publications in the past decade. Because of their primary target, CGs have been promising in their antiviral activities, as the strong activity of these compounds occurs at different stages of the virus species. The main finding on the antiviral activities has stated that these compounds inhibit viral mRNA or protein synthesis, signifying that these drugs target host developments that are important for the viruses to complete an efficacious replication.

Conversely, these mechanisms need to be explored to develop effective drugs with several important advantages such as less risk of resistance and a comprehensive range of action. Here in the current review we mainly focused on identifying the anticancer and antiviral activities of several CGs and we hope that this research may help the researchers to evaluate the anticancer and antiviral potential of CGs in preclinical studies for developing effective drugs.

## 9. Highlights of The Review

Apoptosis is a tightly regulated fundamental process of programmed cell death where the cell finishes its function and automatically undergoes PCD. However, in the case of cancer cells, this mechanism will be disturbed due to the uncontrollable proliferation of cancer cells. Recently, many reports have highlighted that targeting apoptosis through several molecular targets would lead to the discovery of novel anticancer drugs;Cardiac glycosides were used since the ancient years to treat congestive heart diseases, but their anticancer and antiviral activity was found to be novel.Several CGs perturbing apoptosis and autophagy were used to elucidate the mechanism of cell death and are currently in clinical trials;Interestingly CGs act on several targets apart from their primary targets (Na^+^/k^+^-ATPase) such as DNA topoisomerase I and II, anoikis prevention and hypoxia-inducible factors to induce apoptosis;Na^+^/k^+^-ATPase was linked with several signaling pathways such as EMT, Src kinase signaling, p38MAPK or ERK1/2 signaling and PI3K/Akt/mTOR signaling to induce apoptosis and autophagy;Antiviral activities of CGs have demonstrated that CGs can effectively suppress the HIV-1 gene expression, viral protein translation and alters viral pre-mRNA splicing to inhibit several viral diseases such as HIV, HMV, HSV, Ebola, chikungunya and coronavirus;Ultimately, this review provides insights into systematic targeting strategies with recent advancements by using CGs as therapeutic candidates to treat anticancer and antiviral diseases with better efficiency.

## Figures and Tables

**Figure 1 molecules-25-03596-f001:**
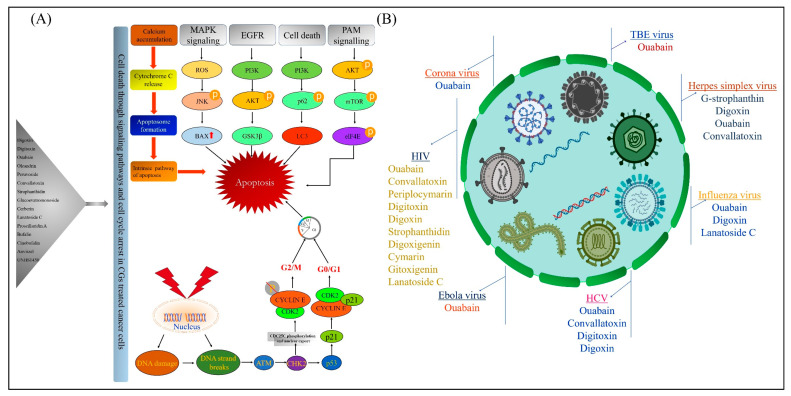
Schematic representation of anticancer and antiviral activities of cardiac glycosides. (**A**) Possible mechanism in human cancers (EGFR—epidermal growth factor receptor; eIF4E—eukaryotic translation initiation factor 4E; PAM—PI3K/AKT/mTOR; PI3K—phosphatidylinositol 3-kinase; mTOR—mechanistic target of rapamycin); (**B**) illustration of potent antiviral diseases (HCV—human cytomegalovirus; HIV—human immunodeficiency virus; HSV—herpes simplex virus; TBE—tick-borne encephalitis virus).

**Table 1 molecules-25-03596-t001:** List of CGs and their mode of action in various cancer cells to induce apoptosis, cell cycle arrest and autophagic cell death.

CGs Name	Mechanism of Action	Reference
Digitoxin	Inhibit general protein synthesis	[62]
	Loss of mitochondrial membrane potential	[51]
	Increase Ca^2+^ uptake	[63]
	Estrogenic receptor antagonist	[64]
	ROS production	[48]
	MAPK pathway mediated apoptosis	[65]
	Topoisomerase I inhibition	[66]
	Decrease in anti-apoptotic proteins Bcl-xL and Bcl-2	[67]
	Increased cytochrome c release and Caspase activation	[68]
	Caspase 9 mediated apoptosis	[69]
	Inhibition of DNA topoisomerases I and II	[70]
	Inhibits p53 synthesis	[71]
Digoxin	Inhibition of Src signaling pathways	[72]
	Inhibition HIF-1alpha synthesis	[40]
	Inhibition of androgen-dependent/independent mechanism	[50]
Ouabain	Inhibits Akt/mTOR signaling pathway	[73]
	Inhibits HIF-1alpha synthesis	[73]
	Androgen-independent apoptosis	[51]
	Autophagic cell death via a JNK-dependent decrease of Bcl-2	[74]
Oleandrin	Inhibition of Wnt/β-catenin signaling	[75]
	Reduction of NF-kB and JNK	[76]
	AP-1 activation	[54]
	Inhibition of 12-O-tetradecanoylphorbol-13-acetate	[77]
	Caspase-3 dependent apoptosis	[78]
	Induces autophagic cell death	[61]
Proscillaridin A	Inhibition of STAT3 activation	[79]
	MYC degradation	[80]
	Apoptosis through calcium-induced DR4 upregulation	[81]
	Cell death through GSK3β activation and alteration of microtubule dynamics	[41]
	Downregulating the expressions of Bcl-xl and MMP2	[82]
Convallatoxin	(mTOR)/p70S6 K signal pathway inhibition	[83]
	p53-independent apoptosis	[30]
	Cell cycle arrest at G0/G1 phase	[8]
UNBS1450	Inhibition of heat shock protein (Hsp70)	[84]
	Augmented permeabilization of lysosomal membrane	[85]
	Blockade of TNF-α/NF-κB signaling pathway	[84]
Lanatoside C	PTEN dependent apoptosis	[86]
	Apoptosis via PKCδ activation	[25]
	Attenuation of Wnt/β-catenin/c-Myc signaling pathway	[87]
	Cell cycle arrest at G2/M phase	[13]
Bufalin	Inhibits topoisomerase II	[88]
	Tiaml mediated apoptosis through activation of Rac1, PAK and JNK pathway.	[89]
	mitochondria-dependent apoptosis through downregulating the miR-221 expression	[90]
	Apoptosis through ROS-dependent RIP1/RIP3/PARP-1 pathways	[81]
	Cell death by upregulation of MiR-203	[91]
	Inhibition of Wnt/ASCL2 expression	[36]
	Inhibits human telomerase reverse transcriptase (hTERT)	[90]
	Apoptosis through downregulation of TGF-β receptors	[92]
	Apoptosis through PTEN/AKT pathways	[93]
Glucoevatromonoside	p53 dependent and independent G2/M arrest	[14]
Cerberin	Inhibition of PI3K/AKT/mTOR signaling	[94]
Digitoxigenin	Inhibition of MMP-2, MMP-9 and p-FAK	[95]
Helleborein	Mitochondrial pathway and caspase-3 dependent apoptosis	[96]
Strophanthidin	Cell death by attenuation of MAPK, PI3K/AKT/mTOR and Wnt/β-Catenin signaling	[12]
Periplocin	Apoptosis by downregulating ATP5A1, EIF5A, ALDH1 and PSMB6	[97]
	Blockade of AKT/ERK signaling	[98]
Peruvoside	Cell cycle arrest at G0/G1 phase	[8]
	Apoptosis via MAPK Wnt/β-catenin and PI3K/AKT/mTOR signaling pathways	[11]
	Caspase and PARP cleavage	[44]
Calotropin	Apoptosis through inhibiting Wnt signaling by increasing casein kinase 1α	[99]
Cymarin	Inhibition HIF-1α synthesis	[40]

**Table 2 molecules-25-03596-t002:** Cardiac glycosides with antiviral activity.

Virus	Cardiac Glycosides	Mechanism of Action Proposed	Reference
Human Cytomegalovirus	digitoxin	Inhibition of NF-κB in CG-treated cells and by modulating human cellular targets associated with hERG	[135]
Human Cytomegalovirus	ouabain	Inhibits viral protein translation	[137]
Human Cytomegalovirus	digoxin	Inhibits viral protein translation	[3]
Human Cytomegalovirus	convallatoxin	Inhibits viral protein translation	[138]
Herpes Simplex Virus	ouabain	Reduces viral protein synthesis	[139]
Herpes Simplex Virus	digoxin	Inhibition of viral gene expression	[140]
Herpes Simplex Virus	digitoxin	Inhibition of viral gene expression	[3]
Herpes Simplex Virus	g-strophanthin	Inhibition of viral gene expression	[3]
Tick-Borne Encephalitis (TBE) Virus	ouabain	Inhibition of to suppress TBE	[141]
Human Cytomegalovirus	digitoxin	Inhibition of NF-κB in CG-treated cells and by modulating human cellular targets associated with hERG	[135]
Human Cytomegalovirus	ouabain	Inhibits viral protein translation	[137]
Human Cytomegalovirus	digoxin	Inhibits viral protein translation	[3]
Human Cytomegalovirus	convallatoxin	Inhibits viral protein translation	[3,138]
Human Cytomegalovirus	digitoxin	Inhibition of NF-κB in CG-treated cells and by modulating human cellular targets associated with hERG	[135]
HIV	convallatoxin, periplocymarin, digitoxin, digoxin, strophanthidin, gitoxigenin diacetate digoxigenin, cymarin, sarmentogenin gitoxin gitoxigenin, strophanthidinic acid lactone acetate, strophanthidin semicarbazide	Inhibition of HIV-1 gene expressionInhibits HIV-1 gene expression by attenuating MEK1/2-ERK1/2 signaling	[142,143]
HIV	digitoxin, lanatoside C, digoxin, ouabain	Alteration of viral pre-RNA splicing	[144]
Chikungunya Virus	digoxin	Inhibition of Src pathway	[145]
Corona Virus	ouabain	Inhibits viral entry through ATP1A1-mediated Src signaling	[140,146]
Ebola Virus	ouabain	Inhibits EBOV replication by around half and the interruption of cellular interacting proteins	[147,148]
Influenza Virus	ouabain, digoxin, lanatoside C	Inhibition of viral proteintranslation	[3,149]

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
