# Peer review of "Anticancer and Antiviral Properties of Cardiac Glycosides: A Review to Explore the Mechanism of Actions"

_molecules, 2020, doi:10.3390/molecules25163596_

Round 1

Reviewer 1 Report

The authors should consider condensing all the known/already reported stuff to a minimum and focus on the new developments/goals achieved instead. For example, the clinical phase 1/2 trials are the same already shown by Schneider et al. (2017) in the same Journal. "More than 60% of the clinically approved drugs for cancer therapy are from the plant sources and around 20-30% of the drugs available in the market were obtained from plants (Perez., 2003)". This is a rather old reference. Please use the figures in David J. Newman and Gordon M. Cragg (2016) Natural Products as Sources of New Drugs from 1981 to 2014. If you 'critically' go into the details you can find 30% NPs, others are derivatives and new formulations, or the NP is used as a lead. But that is not "from plant sources". Table 1 can be omitted since it is not related to the topic. Clinical trials can be omitted (reviewed in 2017, +/- no changes since). The antiviral part should be omitted (another manuscript on this issue was submitted). The article should focus on cancer entities and the putative molecular impact of cardenolides in this specific entity.

The language needs some brush-up. There are typos like "glucoevartomonoside" etc.

Author Response

Q1. The authors should consider condensing all the known/already reported stuff to a minimum and focus on the new developments/goals achieved instead. For example, the clinical phase 1/2 trials are the same already shown by Schneider et al. (2017) in the same Journal.

Response: Thank you for your kind suggestion. We have now improved our manuscript by incorporating the updated information regarding the clinical trials of CGs in the revised manuscript.

Q2. "More than 60% of the clinically approved drugs for cancer therapy are from the plant sources and around 20-30% of the drugs available in the market were obtained from plants (Perez., 2003)".

Response. Thank you for your suggestion. We have changed the reference in the revised manuscript.

Q3. Please use the figures in David J. Newman and Gordon M. Cragg (2016) Natural Products as Sources of New Drugs from 1981 to 2014. If you 'critically' go into the details you can find 30% NPs, others are derivatives and new formulations, or the NP is used as a lead. But that is not "from plant sources".

Response: Thank you for your suggestion. As the manuscript is mainly focusing on the biological properties of CGs (which is also a natural product from plant sources), we originally planned to report only on CGs. However, as suggested by the reviewer, we have used the information from figures of David J. Newman and Gordon M. Cragg (2016) to elaborate the percentage of natural products that have been approved to treat sevral diseases including cancers. Also, the information from the figures was used to discuss the percentages of biological macromolecules, unaltered natural products, botanical drugs (defined mixture), natural product derivatives, synthetic drugs, Synthetic drug (NP pharmacophore)/ mimics of natural products, synthetic drugs (natural product pharmacophores), Vaccines and mimics of natural products in the revised manuscript in the page number 3 (lines 23-32) and page number 4 (lines 1-3) with proper citation.

Q4. Table 1 can be omitted since it is not related to the topic.

Response: Thank you for your suggestion. We have now removed table 1 in the revised manuscript.

Q5. Clinical trials can be omitted (reviewed in 2017, +/- no changes since).

Response: Thank you for your suggestion. Apart from the existing information from Schneider et al., 2017, here we have discussed the completed clinical trials of several CGs. For example, Digitoxin, has cleared the phase II clinical trial for its usage to treat breast cancer on January 31st, 2020. Along with that it has also cleared phase II clinical trials for its usage in treating prostate cancer. Apart from this Digitoxin in combination with Tivantinib has cleared the phase 1 clinical trials to treat solid tumors. A Phase 1B clinical trial for Digitoxin in combination with Trametinib was completed in the year 2018 to treat patients with unresectable or metastatic BRAF wild-type melanoma.

Q6. The antiviral part should be omitted (another manuscript on this issue was submitted). The article should focus on cancer entities and the putative molecular impact of cardenolides in this specific entity.

Response: Thank you for your suggestion. In the present MS we tried to comprehensively explain both the anti-cancer and anti-viral properties of CGs with up-to-date knowledge. As there were not many articles discussing about the mechanism of action of anti-viral activities, we intended to explore the role of CGs in anti-viral activities. Apart from the previous article by Amarelle, L., & Lecuona, E. (2018), in this MS we have discussed several recent reports on anti-viral properties of CGs. For instance, decreasing the influenza virus replication by inhibiting cell protein translational machinery (Amarelle et al., 2019), and inhibition of HIV-1 gene expression through MEK1/2-ERK1/2 signaling (Wong et al., 2018).

References:

  1. Amarelle, L., Katzen, J., Shigemura, M., Welch, L.C., Cajigas, H., Peteranderl, C., Celli, D., Herold, S., Lecuona, E. and Sznajder, J.I., 2019. Cardiac glycosides decrease influenza virus replication by inhibiting cell protein translational machinery. American Journal of Physiology-Lung Cellular and Molecular Physiology, 316(6), pp.L1094-L1106.
  2. Wong RW, Lingwood CA, Ostrowski MA, Cabral T, Cochrane A. Cardiac glycoside/aglycones inhibit HIV-1 gene expression by a mechanism requiring MEK1/2-ERK1/2 signaling. Scientific reports. 2018 Jan 16;8(1):1-7.

Q7. The language needs some brush-up. There are typos like "glucoevartomonoside" etc.

Response: Thank you for you keen observation. We have now corrected the typographical errors and other grammatical mistakes in the revised manuscript.

Reviewer 2 Report

The authors have done a good review on the studies of CGs on cancer cells and cancer models. On section 6, they have briefly discussed clinical trials using four CGs on cancer patients. Finally, they briefly summarized anti-viral activities of CGs and potential mechanisms of action.

Some minor issues are:

  1. Line 349. One of the markers of ICD is ecto-expression of calreticulin, not the expression of calreticulin itself. Calreticulin is located in storage compartments associated with the endoplasmic reticulum and is considered an ER resident protein. When cells are dying, it may be presented to the cell surface (called ecto-expression) and acts as a danger signal, thus one of markers for ICD.  Please make a correction.
  2. Figure 1A. due to bad color shading, some words are not easily visible. Please make adjustments.
  3. Figure 1 legend. When tumor is malignant, it is called cancer. Thus, “malignant cancer” is a redundant expression.
  4. Page 12, line 441. Typo: Trail.
  5. Based on the information on Section 6, four CGs are in clinical trial. Therefore, In the Abstract, line 27, it is “in clinical trials for…” not “in preclinical trials for…”.
  6. References: some of them are missing volume, page numbers or article number. These references include, #12, 32, 63, 67, 130, 135, 142, and 148.
  7. Two references: In these cases, they are e-journals and should cite the article number, not the page numbers. Ref #50, article number is 17009.  Ref #54, the article number is 29721.  Please correct them.
  8. The citation format in the main text is different from the standard one used for this journal.

Author Response

The authors have done a good review on the studies of CGs on cancer cells and cancer models. On section 6, they have briefly discussed clinical trials using four CGs on cancer patients. Finally, they briefly summarized anti-viral activities of CGs and potential mechanisms of action.

Response: Thank you very much for nicely summarising the manuscript.

Some minor issues are:

Q1. Line 349. One of the markers of ICD is ecto-expression of calreticulin, not the expression of calreticulin itself. Calreticulin is located in storage compartments associated with the endoplasmic reticulum and is considered an ER resident protein. When cells are dying, it may be presented to the cell surface (called ecto-expression) and acts as a danger signal, thus one of markers for ICD. Please make a correction.

Response: Thank you for the observation. The sentence was now modified as per the suggestion.

Q2. Figure 1A. due to bad color shading, some words are not easily visible. Please make adjustments.

Response: The figure colours were now adjusted for easy visibility.

Q3. Figure 1 legend. When tumor is malignant, it is called cancer. Thus, “malignant cancer” is a redundant expression.

Response: Thank you for your suggestion. The sentence is modified as “cancer” in the revised manuscript.

Q4. Page 12, line 441. Typo: Trail.

Response: Thank you for your keen observation throughout the manuscript. The word is now modified as “Trial” in the revised manuscript.

Q5. Based on the information on Section 6, four CGs are in clinical trial. Therefore, In the Abstract, line 27, it is “in clinical trials for…” not “in preclinical trials for…”.

Response: Thank you for your observation. As per the suggestion the sentence is now modified as “clinical trials” in the revised manuscript.

Q6. References: some of them are missing volume, page numbers or article number. These references include, #12, 32, 63, 67, 130, 135, 142, and 148.

Response: Thank you for your observation. We have included the volume numbers in the revised manuscript.

Q7. Two references: In these cases, they are e-journals and should cite the article number, not the page numbers. Ref #50, article number is 17009. Ref #54, the article number is 29721. Please correct them.

Response: Thank you for your valuable suggestion. We have modified both references as per your suggestion in the revised manuscript.

Q8. The citation format in the main text is different from the standard one used for this journal.

Response: Thank you for your suggestion. The citation format and reference style have now been changed as per the journal guidelines.